# All-printed stretchable corneal sensor on soft contact lenses for noninvasive and painless ocular electrodiagnosis

Kyunghun Kim[1,8], Ho Joong Kim[2,8], Haozhe Zhang[3,8], Woohyun Park [4], Dawn Meyer[5], Min Ku Kim [1], Bongjoong Kim[4], Heun Park[1], Baoxing Xu [3✉], Pete Kollbaum [5✉], Bryan W. Boudouris [2,6✉] & Chi Hwan Lee [1,4,7✉]

Electroretinogram examinations serve as routine clinical procedures in ophthalmology for the diagnosis and management of many ocular diseases. However, the rigid form factor of current corneal sensors produces a mismatch with the soft, curvilinear, and exceptionally sensitive human cornea, which typically requires the use of topical anesthesia and a speculum for pain management and safety. Here we report a design of an all-printed stretchable corneal sensor built on commercially-available disposable soft contact lenses that can intimately and non-invasively interface with the corneal surface of human eyes. The corneal sensor is integrated with soft contact lenses via an electrochemical anchoring mechanism in a seamless manner that ensures its mechanical and chemical reliability. Thus, the resulting device enables the high-fidelity recording of full-field electroretinogram signals in human eyes without the need of topical anesthesia or a speculum. The device, superior to clinical standards in terms of signal quality and comfortability, is expected to address unmet clinical needs in the field of ocular electrodiagnosis.

[1] Weldon School of Biomedical Engineering, Purdue University, West Lafayette, IN, USA. [2] Charles D. Davidson School of Chemical Engineering, Purdue University, West Lafayette, IN, USA. [3] Department of Mechanical and Aerospace Engineering, University of Virginia, Charlottesville, VA, USA. [4] School of Mechanical Engineering, Purdue University, West Lafayette, IN, USA. [5] School of Optometry, Indiana University, Bloomington, IN, USA. [6] Department of Chemistry, Purdue University, West Lafayette, IN, USA. [7] School of Materials Engineering, Purdue University, West Lafayette, IN, USA. [8] These authors contributed equally: Kyunghun Kim, Ho Joong Kim, Haozhe Zhang. ✉email: bx4c@virginia.edu; kollbaum@indiana.edu; boudouris@purdue.edu; lee2270@purdue.edu

Electrophysiological activity of the retina in response to a light stimulus, known as an electroretinogram (ERG), is recorded at the corneal surface in ophthalmic examinations for the diagnosis or early detection of many ocular diseases such as glaucoma, retinitis pigmentosa, diabetic retinopathy, retinoschisis/detachment, and other congenital degenerations[1–3]. The measurement of ERG signals occurs by contacting a recording electrode directly with either (1) the corneal surface or (2) the bulbar conjunctiva while placing a grounding electrode and a reference electrode on the earlobe and forehead, respectively[4,5]. The current gold-standard method for measuring ERG signals involves the use of contact lens-type devices (e.g., the ERG-Jet lens) that facilitate direct contact to the corneal surface and thereby enable the recording of ERG signals with relatively higher amplitudes than conjunctival electrodes[6]. However, these devices consist of a thick, rigid contact lens with non-optimal geometries (in particular, anteriorly protruding bumps and large outer curvature for human eyes), resulting in discomfort to both the cornea and eyelid despite ocular topical anesthesia. This discomfort is not easily tolerated (especially by children and adults with poor cooperation), and thereby general anesthesia or sedation is often required for these patients[7,8]. In cases of patient refusal of anesthesia or sedation, hook-type conjunctival devices [e.g., the Dawson Trick Litzkow (DTL) fiber] can alternatively be used. However, the signal quality with these devices is significantly compromised (e.g., <46%) due to its far distance from the cornea, limiting the interpretability of the obtained data[9,10]. Newer versions of contact lens-type devices (e.g., the Burian-Allen lens) include a built-in speculum that prevents blinking, and, therefore, enhance the safety and ease of use of these devices from the practitioner standpoint[11]. However, the bulky size of the built-in speculum limits its use only on sedated patients due to severe discomfort. Its use, therefore, is primarily reserved for rare clinical conditions that demand a long-term recording of ERG responses over several hours[12,13]. Moreover, these devices remain expensive, and, therefore, they are often reused multiple times across different patients. This reuse requires a thorough disinfection process in which the practitioner may lack complete confidence, especially with ongoing issues of easily transferable viruses [e.g., the Coronavirus disease 2019 (COVID-19)].

Recent technological advances have led to the development of industrial-grade smart contact lenses, such as the Sensimed TriggerFish lens and the Google smart contact lenses. These devices allow for (1) the continuous monitoring of intraocular pressure (IOP) or biomarkers (e.g., glucose) in tear at the corneal surface and (2) the wireless transmission of the data to the wearer through the use of an integrated circuit (IC) chip[14]. However, the IC chip embedded in these devices is at least >3-fold thicker and >75,000-fold stiffer than a typical soft contact lens (SCL), which results in user discomfort and the risk of corneal hypoxia, especially if worn for a long period of time. Other side effects have been also reported, including foreign body sensation, eye pain, superficial punctate keratitis, corneal epithelial defects, and conjunctival erythema[15]. More recently, several ongoing research endeavors have helped enable the successful fabrication of a range of flexible sensors on a custom-built contact lens made from several polymers (e.g., hydrogel silicones, Parylene-C, or SU8 resins) and functional nanomaterials (e.g., graphene and metallic nanowires)[16–19]. These newer devices have shown some initial success at the laboratory scale, but their practical application in human eyes remains impeded due to the lack of mechanical reliability (for lens handling, fitting, cleaning, and inadvertent eye rubbing), chemical stability (for long-term lens storage and multiple disinfection cycles), and oxygen transmissibility. Moreover, the custom-built contact lenses used in these devices still suffer from limited wettability or achieving ergonomic curvature, which may affect their long-term wearability for the human eye.

Here, we report an innovative strategy that involves the direct-in-writing (DIW) of a highly stretchable ERG corneal sensor on various types of commercial disposable SCLs that offer excellent biocompatibility, softness [mechanical modulus ($E$) = 0.2–2 MPa], transparency (~100%), oxygen transmissibility (10–200 Dk/$t$), wettability (water content = 30–80%), and are also able to fit a variety of corneal shapes (8.3–9.0 mm base curve radii)[20,21]. Being placed on a commercial SCL, which conforms to an arbitrary corneal shape, the resulting device provides unique capabilities to (1) capture high-fidelity ERG signals in human eyes without the use of corneal anesthesia or a speculum, (2) fit well for an arbitrary size or shape of human eyes, and (3) be less decentered on the eye by >10-fold compared to the ERG-Jet lens without scratching the corneal surface. As schematically illustrated in Fig. 1a, the device includes a circular serpentine trace of conduction paths located at the outer peripheral edge of a SCL, allowing light to pass unobstructed through the center lens region. In this design scheme, wireless ERG recording is unnecessary because most of the clinical ERG examinations are routine in-office procedures and typically occur within no more than 30 min in a clinic in the presence of a sophisticated light stimulator (e.g., a Ganzfeld stimulator). Instead, the device is connected to an external data acquisition system via a custom-built thin connection wire that is exceptionally stretchable (up to 350%) and lightweight (~1.4 mg cm$^{-1}$) to minimize the effects of blinking and eye rotational movements (e.g., on average ~±4 mm) on signal quality. This connection wire is >5-fold thinner, >6-fold lighter, and >3000-fold softer than a conventional lead wire that is also used for the ERG-Jet lens.

Consequently, the device offers significant advantages over both the commercially available clinical vision technologies and the recently explored/in-development smart contact lenses. (1) The device consists of intrinsically stretchable polymers, of which the stacked layers remain at least 7-fold thinner, 2-fold softer, and 10-fold more stretchable compared to commercial SCLs. The device is also >25-fold thinner, >3-fold lighter, and >2000-fold softer than the ERG-Jet lens. (2) The device is directly printed on various types of commercial SCLs without substantially altering the intrinsic lens properties. It, therefore, offers excellent wettability, biocompatibility, and oxygen transmissibility, compared to bare SCLs. (3) The device is monolithically bonded to commercial SCLs through an electrochemical anchoring mechanism to provide sufficient mechanical and chemical reliability even under harsh environmental conditions, including overstretching, temperature cycling between 30 and 80 °C, and multiple dehydrations in ambient conditions for at least 5 h each. In our preclinical tests, the device established an optimally aligned conformal interface with the corneal anterior surface of a human eye similar to bare SCLs. These aspects allowed the device to provide significantly improved measurement accuracy and comfort without the use of topical corneal anesthesia or a speculum (as is typically used in current ophthalmic examinations despite its adverse effects)[22].

## Results

**Basic design, layout, and fabrication strategy.** Figure 1b presents schematic representations of the overall device design. The corneal sensor is configured into a thin, narrow serpentine trace (250 μm wide × 10 μm thick × 66 mm long) and positioned on the inner surface of a commercial disposable SCL facing the corneal surface. A conductive biocompatible polymer, poly(3,4-ethylenedioxythiophene) (PEDOT) doped with tosylate[23–25], is electrochemically printed around the entire outer surface of the corneal sensor in order to provide a thin encapsulation layer and

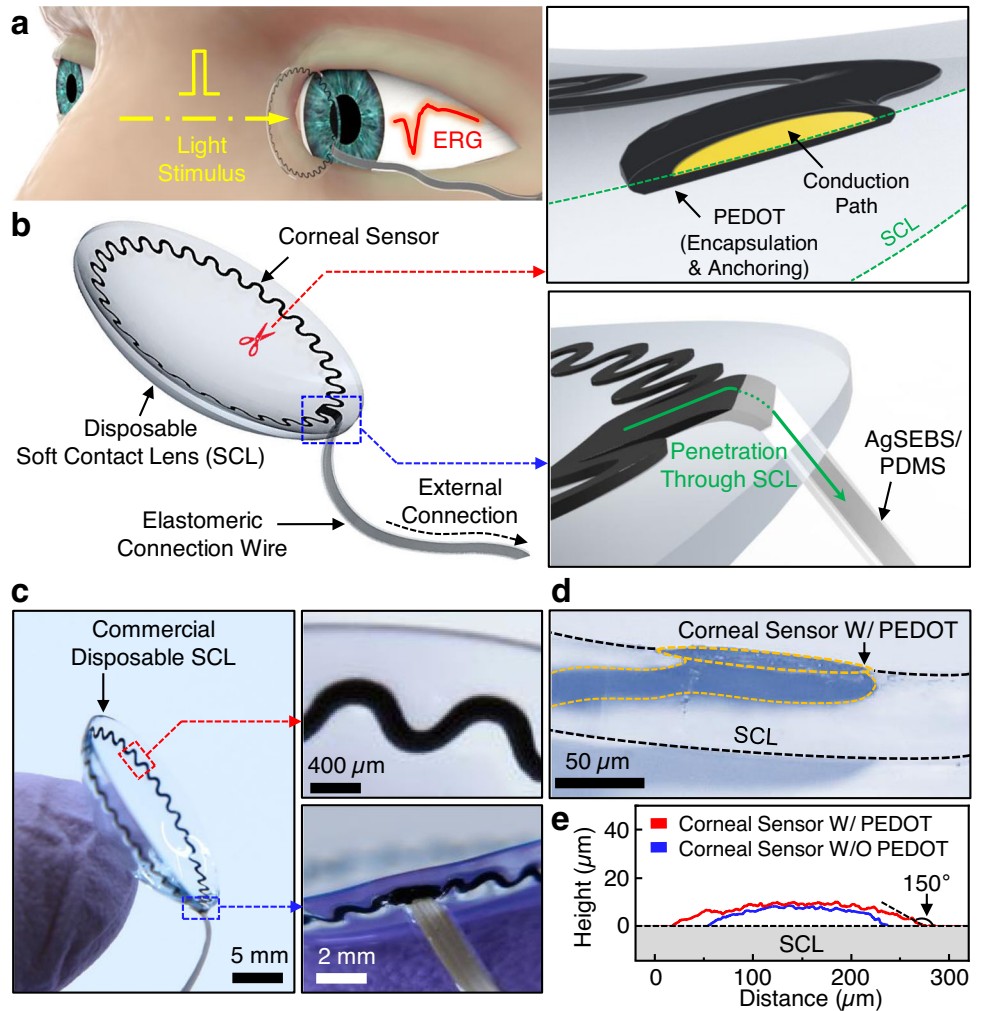

**Fig. 1 Schematic illustrations and photographs of the all-printed corneal sensor for electroretinogram (ERG) recording. a** Schematic illustration of ERG recording in response to a light stimulus from a human eye using the corneal sensor. **b** Schematic illustrations of the corneal sensor, with inset images highlighting the embedded encapsulation and anchoring layers (top panel) and its seamless integration with the connection wire (bottom panel). **c** Photographs of the corneal sensor, with inset images highlighting the embedded serpentine layout (top panel) and its seamless integration with the connection wire (bottom panel). **d** Cross-sectional microscope image of the corneal sensor. **e** Surface topology of the corneal sensor with and without a PEDOT layer.

promote anchoring to the SCL (Fig. 1b, top inset image). The corneal sensor is monolithically linked to an elastomeric connection wire (1 mm wide × 120 μm thick × >5 cm long) that comprises custom-formulated elastomers, including silver flake-filled polystyrene-*b*-poly(ethylene-*co*-butylene)-*b*-polystyrene (AgSEBS) and fumed silica nanoparticle-filled poly-dimethylsiloxane (PDMS)[26,27]. Here, the connection wire penetrates through the SCL for seamless integration (Fig. 1b, bottom inset image). Figure 1c shows the overall size and design configuration of the resulting device built upon a disposable SCL (ACUVUE Oasys, Johnson & Johnson; center thickness of 70 μm).

Supplementary Figure S1 shows schematic illustrations of the entire procedure for fabricating the corneal sensor. The fabrication begins with an automated nozzle injection tool equipped on a three-axis computer-controlled translation stage (Nordson EFD, resolution: 1 μm, repeatability: ±3 μm). This tool allows for the DIW of elastomeric inks (e.g., the formulated AgSEBS and PDMS) on a glass substrate coated with a water-soluble polyvinyl alcohol (PVA) layer. This printing technique provides the versatility to write multiple layers of linear and

curvilinear traces uniformly at the microscale (>100 μm in width and >10 μm in thickness) in a series of pre-programmed steps, enabling batch production (>10 units per print). The real-time demonstration of this automated batch processing is shown in Supplementary Movie S1. The next step involves removing the water-soluble PVA layer with deionized (DI) water, followed by electroplating the conduction path (i.e., AgSEBS) with gold (Au) not only to promote electrical conductivity but also to enhance scratch resistance and chemical stability within aqueous media[28]. The as-printed corneal sensor is then transferred to the inner surface of a SCL, while the elastomeric connection wire is inserted out through the SCL. The serpentine trace of the corneal sensor is stretched when contacted to the curvilinear surface of the SCL until it accommodates the interfacial stress, thereby avoiding any surface discontinuity[29]. The next step involves an electrochemical polymerization of 3,4-ethylenedioxythiophene (EDOT) to form a thin PEDOT layer over the Au-coated surface (Supplementary Figure S2). Finally, the resulting device is thoroughly washed with a preservative-free saline solution, followed by an overnight sterilization process with a

commercial disinfection hydrogen peroxide ($H_2O_2$) solution. Details of the materials and fabrication procedures are also shown in the "Methods" section.

The cross-sectional microscope image in Fig. 1d shows that the electrochemically grown PEDOT layer conformed to the surface of the corneal sensor and seamlessly penetrated the SCL. Figure 1e shows the surface topology of the corneal sensor with (red line) and without (blue line) a PEDOT layer, of which the electrochemical processing time was fixed at 4 min. The results indicate that the peak heights remained <10 μm, while the formation of the PEDOT layer occurred predominantly at the edge of the corneal sensor due to uneven current distribution across the round surface[30]. Consequently, a gradual taper angle of ≤30° was created at the edge of the PEDOT layer, offering enhanced conformal contact to the corneal surface. These features are important to minimize irritation to the cornea while reducing the edge stress[31]. The corresponding results of the gradually tapered PEDOT layer as a function of varied electrochemical processing time ranging from 1 to 4 min are summarized in Supplementary Figure S3. These observations were reproducible across different types (e.g., materials, water content, and ionicity) of commercial disposable SCLs (Supplementary Figure S4).

**Mechanical and chemical characterization**. The experimental results in Fig. 2a show that the device provides a similar modulus ($E = 790 \pm 140$ kPa) to a bare control SCL (blue line; ACUVUE Oasys, Johnson & Johnson), while the corneal sensor itself (without the SCL) provides >2-fold lower modulus (green line; $E = 374 \pm 47$ kPa). The low modulus of the bare corneal sensor allows its addition to a SCL without substantially altering the mechanical properties of the SCL, which would otherwise perform differently. In fact, the bare corneal sensor is at least 7-fold thinner than the SCL (>70 μm thick) and takes only 8% of the total surface area of the SCL on the peripheral edge to further minimize the effect on overall lens performance. For instance, the corneal sensor itself was stretched without failure even after the SCL was torn into two pieces at the maximum strain of ~100% (Supplementary Figure S5). The results also show that the monolithically integrated elastomeric connection wire (purple line; $E = 420 \pm 41$ kPa) is virtually as soft as the bare corneal sensor, and, therefore, should have minimal effect on blinking or eye movements. The connection wire was stretched up to 350% prior to its mechanical failure, while the relative change in resistance ($\Delta R/R_0$) remained <2.7 (Fig. 2b). A representative cross-sectional scanning electron microscope (SEM) image of the connection wire is shown in Supplementary Figure S6. The mechanical and electrical properties of the connection wire were negligibly changed after >1500 cycles of stretching at 50% (Fig. 2c, top panel), resulting in well-maintained electrochemical impedance of the corneal sensor (Fig. 2c, bottom panel). The real-time demonstration of stretching the corneal sensor is shown in Supplementary Movie S2. These assessments were consistent with experimental observations by twisting the connection wire up to 1440° for >1500 cycles (Fig. 2d). In addition, its electrochemical impedance remained sufficiently low at $18.2 \pm 3.8$ Ω even against tapping, swinging, and spinning of the connection wire (Supplementary Figure S7), implying that the effect of motion artifacts (e.g., blinking or eye movements) on signal quality is insignificant.

Figure 2e shows experimental measurements for the frequency-dependent electrochemical impedance of the corneal sensor with (red line) and without (blue line) a PEDOT layer in a solution of 1× phosphate-buffered saline (PBS, pH = 7.4), by comparison with widely used clinical standards such as the ERG-Jet lens (LKC Technologies; center lens thickness of 500 μm) and the DTL fiber

(Diagnosys; seven times interwoven fiber with the outer diameter of 0.8 mm) (Supplementary Figure S8)[32]. The corneal sensor with a PEDOT layer showed the lowest impedance (<100 Ω) among the three devices within the typical frequency range of ERG recordings in human eyes (gray highlighted area; 0.3–300 Hz)[33], which would therefore give rise to the high signal-to-noise ratio. These observations were reproducible from device to device (Supplementary Figure S9). The impedance of the corneal sensor was nearly unchanged over >1000 cycles of folding and scrubbing (Fig. 2f) and after 30 days of immersion in several aqueous media, such as lens cleaning solution (Sensitive Eyes® saline solution, Bausch & Lomb), PBS (pH = 7.4; Gibco), and artificial tear (Refresh Tears® lubricant eye drops, Allergan) at 100 Hz (Fig. 2g). The real-time demonstration of folding and scrubbing the corneal sensor is shown in Supplementary Movie S3. The corresponding results of the impedance as a function of frequency are summarized in Supplementary Figure S10. Figure 2h confirms that the impedance was negligibly changed throughout multiple disinfection cycles (>5 times) by immersing the corneal sensor in a cleansing kit (inset image) filled with a 3% $H_2O_2$ formula (ClearCare®, Alcon) for 12 h each. During these disinfection cycles, no evidence of visual changes in the appearance of the corneal sensor was observed (Supplementary Figure S11). The impedance was also well maintained under other harsh environmental conditions, such as temperature cycling between 30 and 80 °C and multiple dehydrations in ambient conditions for at least 5 h each (Supplementary Figure S12). The impedance was slightly decreased at a high temperature >60 °C.

Time-dependent cytotoxicity of the corneal sensor to human corneal cell lines is an essential consideration to identify any adverse responses in vitro[34,35]. Figure 2i shows a cell viability assay of human corneal epithelial cells (HCEpiCs) that were seeded on the surface of the corneal sensor with (red bars) and without (blue bars) a PEDOT layer in a culture medium (EpiGRO™ Human Ocular Epithelia Complete Media, MilliporeSigma) at 37.5 °C. For all cases, the cell viabilities were retained over 95% throughout the entire assay period (24 h) without substantial differences relative to bare control cells (green bars). The results imply that the corneal sensor would provide little risk for the development of corneal inflammation during ERG examinations. The results also confirm that there was no residual EDOT present after the washing and disinfecting processes. Details of the cell culture and associated characterization procedures are shown in the "Methods" section.

**Mechanics analysis under various loading conditions**. The low mechanical modulus of the corneal sensor reduces the risk for mechanical failure against various loading conditions required for lens handling, cleaning, storage, and fitting. Figure 3 summarizes experimental (left column) and finite element analysis (FEA; middle column) results of the corneal sensor under four different loading conditions: (a) flipping, (b) folding, (c) stretching (up to 40%), and (d) expanding (up to 10%). For control comparisons, the corresponding FEA results for a bare corneal sensor (without the SCL) are shown in the right column of Fig. 3. The results show that the maximum principal strain ($\varepsilon_{max}$) of the corneal sensor remained lower than ~10% under these loading conditions. For example, when completely flipped over, the corneal sensor experienced little deformation with the maximum strain of <1% (Fig. 3a). When folded in half along the symmetric axis, the maximum strain (<10%) was concentrated at the folding line of the corneal sensor (Fig. 3b). When stretched uniaxially and expanded uniformly, the results consistently showed that the maximum strain remained <10% (Fig. 3c, d). For all cases, the maximum strains of the corneal sensor were higher than the bare

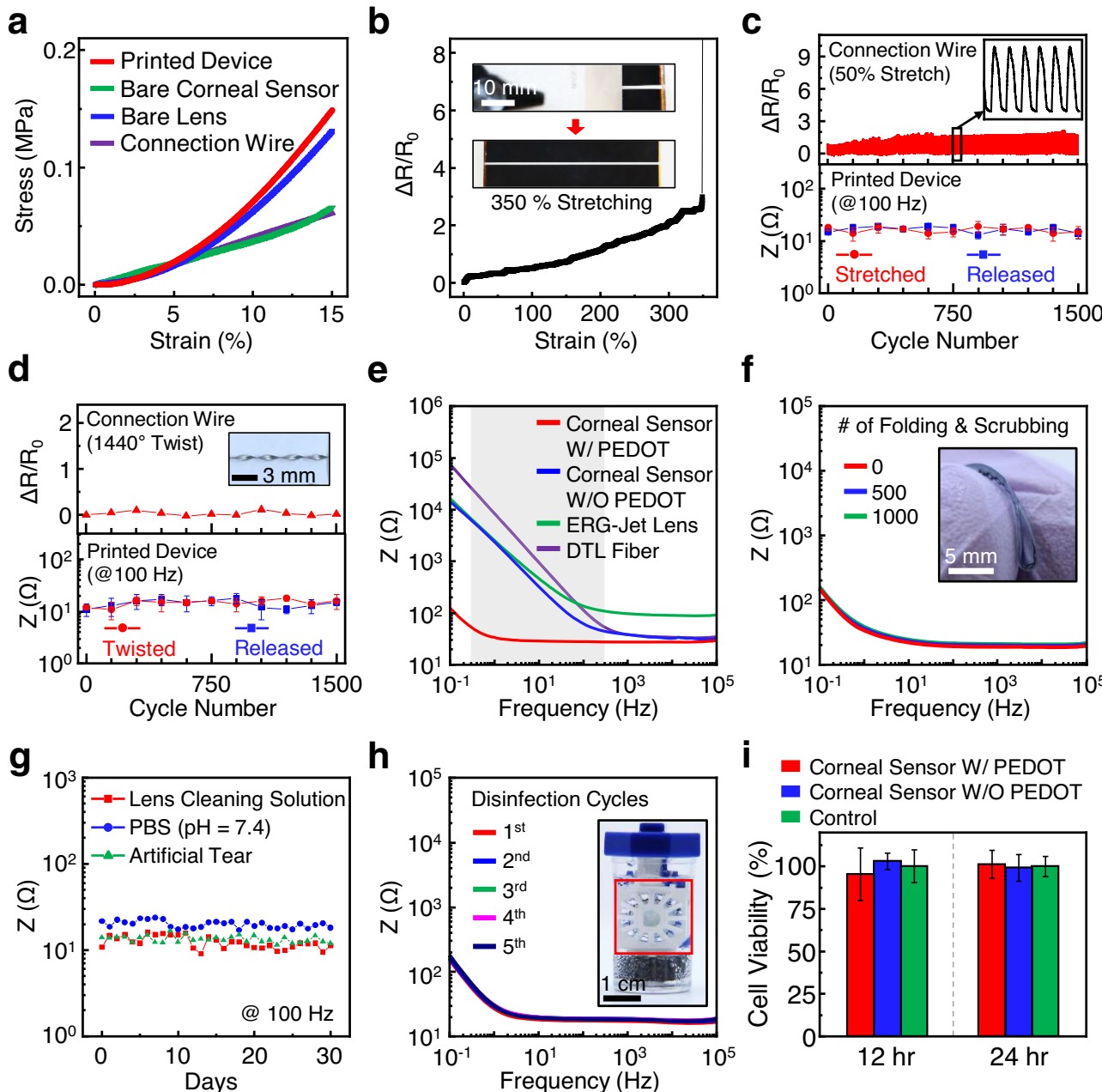

**Fig. 2 Characterizations of the corneal sensor. a** Average stress–strain curves for the corneal sensor, the bare corneal sensor (without the SCL), the bare SCL, and the connection wire. **b** Relative change in resistance ($\Delta R/R_0$) of the connection wire under stretching up to 350%. The inset images show the stretched wire. **c** $\Delta R/R_0$ of the connection wire under 1500 cycles of stretching at 50% (top panel) and the consequent change in the electrochemical impedance of the corneal sensor at every 150 cycles (bottom panel). **d** $\Delta R/R_0$ of the connection wire under 1500 cycles of twisting up to 1440° (top panel) and the consequent change in the electrochemical impedance of the corneal sensor at every 150 cycles (bottom panel). **e** Electrochemical impedance of the corneal sensor as a function of frequency by comparisons with current clinical standards. **f** Electrochemical impedance of the corneal sensor under 1000 cycles of folding and scrubbing. **g** Electrochemical impedance of the corneal sensor immersing in several aqueous media for 30 days. **h** Electrochemical impedance of the corneal sensor for five cycles of disinfection processes. The inset image shows a commercial cleansing kit filled with a 3% $H_2O_2$ formula. **i** Cell viability assay of human corneal epithelial cells (HCEpiCs) seeded on the corneal sensor with and without a PEDOT layer by comparison with bare control cells ($n = 5$).

corneal sensor (without the SCL) by several factors, indicating that the mechanical deformations occurred primarily on the SCL rather than the corneal sensor itself. These findings also imply that the effect of the bare corneal sensor on the intrinsic mechanical properties of the SCL was insignificant.

**Real-time ERG recording in human eyes**. To demonstrate the feasibility and measurement validity in the human eye, a pilot

evaluation of the corneal sensor was conducted by clinical research-trained personnel on a healthy adult participant (a 45-year-old male) who had no history of ocular disease. Prior to and following ERG recordings, several pre-examination data were acquired including visual acuity, participant-reported comfort ratings, ocular coherence tomography (OCT) (Visante, Zeiss), and slit lamp biomicroscopic (SL120, Zeiss) measures of ocular health and lens fit. The slit lamp biomicroscopic measures were

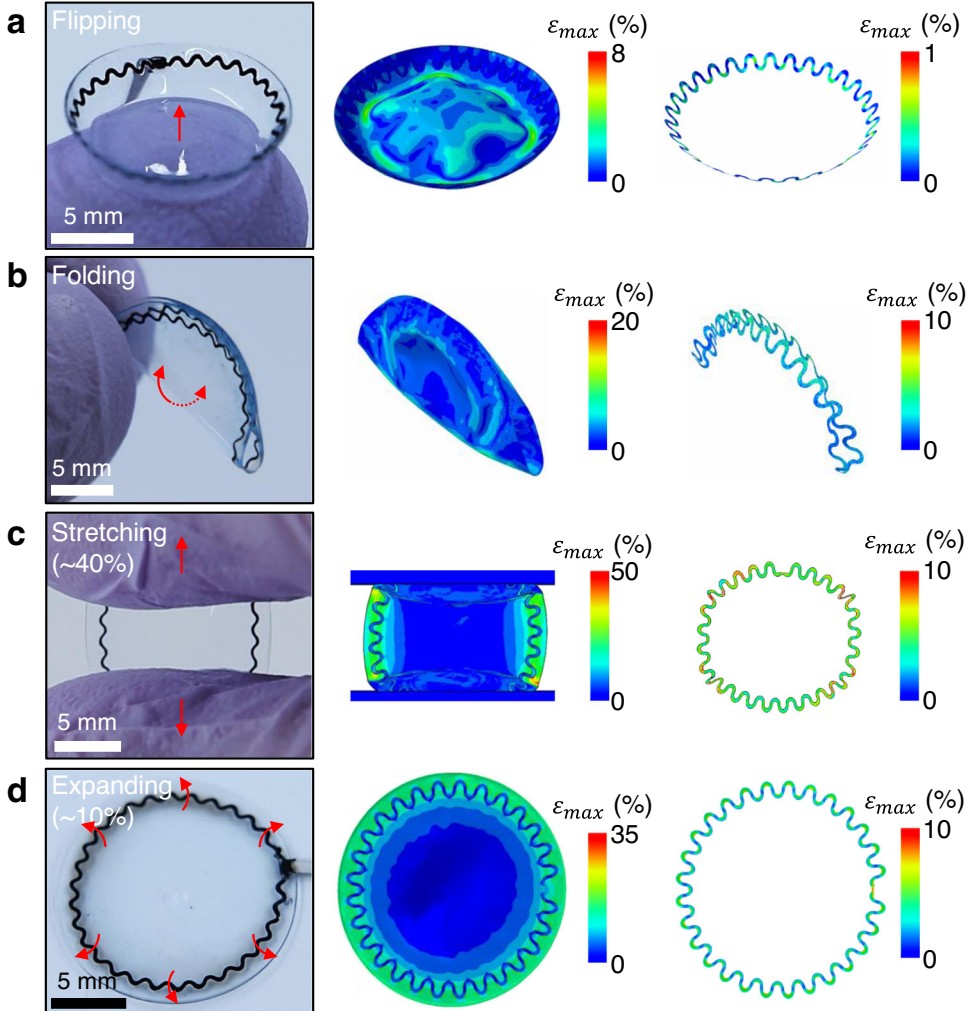

**Fig. 3 Mechanics analysis under various loading conditions.** Photographs (left column) and finite element analysis (FEA) results (middle column) of the corneal sensor under four different loading conditions: **a** flipping, **b** folding, **c** stretching (up to 40%), and **d** expanding (up to 10%). The right column shows the corresponding FEA results of the bare corneal sensor without the SCL.

acquired with normal white lights as well as sodium fluorescein installation (1 mg; Fluorets ophthalmic strips, Bausch and Lomb), which highlights any area of damage on the corneal epithelial surface. The ERG recordings were acquired with the corneal sensor and two comparator controls: the ERG-Jet lens and DTL fiber[32]. The Burian-Allen ERG electrode was not included in this study due to its high discomfort and low tolerance experienced by the participant[5].

Figure 4a demonstrates the process of ERG recordings where the participant was seated upright in front of a Ganzfeld stimulator (RETI-port/scan 21, Roland Consult), which generated a series of short-wavelength stimuli in low luminance conditions (~0.001 cd m$^{-2}$)[33]. The corneal sensor was inserted on the left eye of the participant without topical anesthesia or a speculum, and the participant was asked to blink normally. Biomicroscopic examination revealed a mobile, well-fit lens with good centration, coverage, and movement (Fig. 4b, top panel). Both the corneal sensor and the connection wire had little impact on blinking or eye movements with the gaze angle of up to ±40° (which most commonly occur), resulting in the lens movements being as smooth as a bare SCL (Supplementary Movie S4). In a temporal gaze larger than these angles (which is not common in normal eye movements), the connection wire temporarily adhered to the wet conjunctival epithelium and prevent the SCL from continuing to

rotate with the eye. This impact could be minimized by the selective inferior placement of the connection wire. For control comparisons, both the ERG-Jet lens and the DTL fiber were tested on the same eye of the participant to maximize comparison of the techniques and eliminate the effect of different eye sizes and shape on the ERG signals (Fig. 4b, middle and bottom panels, respectively)[36]. Moreover, the ERG signals are independent of the size and shape of the human eye, and thereby the ERG measurement requires no calibration among different subjects because initial participant measurement data are used as a reference baseline[5]. Prior to the implementation of the ERG-Jet lens, the eye was anesthetized with one drop of 0.5% proparacaine hydrochloride and then moistened with 0.5% methylcellulose in order to reduce discomfort. No speculum was used, but the participant was not able to fully blink with the lens in due to the four built-in anterior bumps (yellow circles; 1.5 mm wide and 2.5 mm long each) preventing complete eyelid closure[6]. The base curve radius of the lens was 7.9 mm, which is slightly flat relative to the central corneal curvature of the participant (7.7 mm). Care was taken to ensure adequate alignment of the lens over the pupil center during measurements, but as is typical for any rigid lens, it moved ~1–1.5 mm on the eye[37]. As the last control measure, the DTL fiber was gently placed across the bulbar conjunctiva above the lower eyelid without topical anesthesia[38]. Figure 4c shows

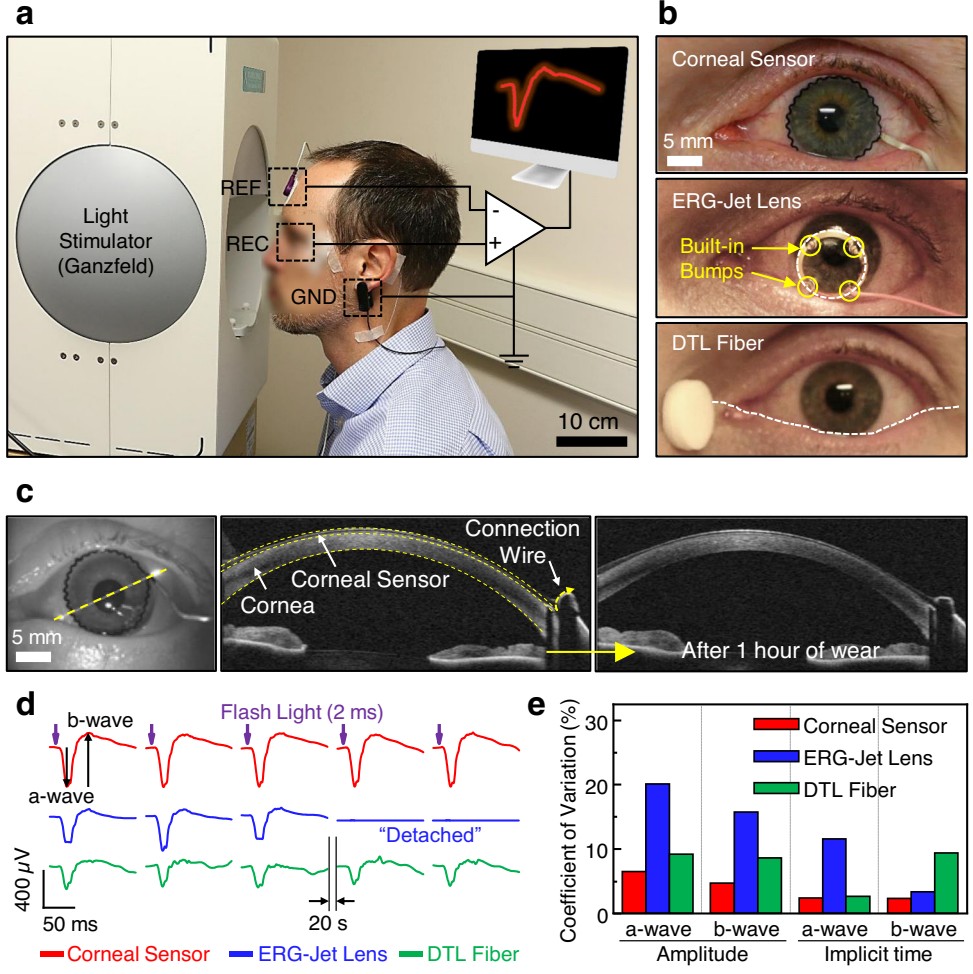

**Fig. 4 Real-time ERG recordings in a human eye. a** Measurement setting for ERG recordings using a Ganzfeld stimulator in scotopic conditions.
**b** Photographs of the eye worn with the corneal sensor (top panel) by comparisons with the ERG-Jet lens (middle panel) and DTL fiber (bottom panel). The
yellow circles denote the four built-in bumps of the ERG-Jet lens. The white dotted line denotes the outer trace of the DTL fiber. **c** Photograph (left inset)
and the corresponding anterior segment ocular coherence tomography (AS-OCT) images of the corneal sensor worn on the cornea upon insertion (middle
inset) and after 1 h of the wear (right inset). **d** Full-field ERG signals acquired from the three different devices under the light intensity of 10.0 cd·s m$^{-2}$.
**e** Coefficient of variation (CV) of the amplitudes and implicit times of the a and b waves extracted. The number of recordings with each device was ten
(corneal sensor), eight (ERG-Jet lens), and nine (DTL fiber) recordings, respectively.

representative images of anterior segment OCT (AS-OCT),
confirming the conformational alignment of the corneal sensor
with the cornea upon insertion (middle inset) and after 1 h of the
wear (right inset). The corresponding AS-OCT images using the
ERG-Jet lens could not be acquired due to the unstable contact of
the device to the anterior corneal surface and the built-in bumps
preventing axial instrument focus. Participant-reported comfort
was assessed prior to and during each use of the devices
employing a simple 100-point numeric scale[35,39], where a rating
of "1" represented extremely uncomfortable/intolerable and a
rating of "100" perfectly comfortable/not noticeable at all. Prior to
testing, the participant provided a rating of 95 with his habitual
SCLs (Dailies Aqua Comfort Plus, Alcon), compared to a rating
of 98 with no contact lens at all. The DTL fiber was given a rating
of 88 versus the corneal sensor of 86 (both without topical
anesthesia). Alternatively, the ERG-Jet lens with topical anesthe-
sia was given a numeric rating of 42.

These pilot tests revealed the following important findings: (1)
The corneal sensor conformed well to the cornea when on the
eye, whereas the ERG-Jet lens created a gap between the corneal
sensor and the cornea of ~500 μm (the sum of the lens thickness
and a thick aqueous tear layer between the posterior lens and

anterior cornea) due to its relatively flat curvature. (2) The
corneal sensor remained centered on the cornea, whereas the
ERG-Jet lens was systematically decentered. (3) The corneal
sensor was rated to be much more comfortable and easier to use
than the ERG-Jet lens even without the use of topical anesthesia,
and in line with the DTL fiber that was not in contact with the
cornea. (4) The external connection wire of the corneal sensor
was thin (120 μm thick), lightweight (~1.4 mg cm$^{-1}$), and
sufficiently soft ($E = 420$; kPa) enough to avoid any interruption
from blinking and eye movements (Supplementary Movie S5).
On the other hand, the polyvinyl chloride-coated lead cable of the
ERG-Jet lens was considerably thick (0.6 mm diameter), heavy
(8.6 mg cm$^{-1}$), and stiff ($E = 1.3$ GPa), making it difficult to align
the lens to the pupil center and capture consistent ERG signals
(Supplementary Movie S6). Details of the technical specifications
of these devices are compared in Supplementary Table S1. These
experimental observations obtained with the ERG-Jet lens and the
DTL fiber are consistent with previous reports[11,32,40].

Following application of the devices and prior to ERG
recording, the participant was asked to sit in the low luminance
room (<0.001 cd m$^{-2}$) for at least 20 min. The participant was
then asked to gaze at a fixation spot inside the Ganzfeld bowl to

maintain a constant amount of light transmission to the retina and minimize interference that could be generated upon ocular movements[41]. The participant's head was kept within 5 cm from the Ganzfeld bowl opening. The pupil size was continuously monitored using an infrared (IR) camera inside the Ganzfeld dome (Supplementary Figure S13). Figure 4d shows representative ERG signals obtained by consecutively illuminating a white flashing stimulus (10.0 cd·m$^{-2}$) for 2 ms at the interval of 20 s to allow the pupil to fully dilate again. For all devices, the results showed typical scotopic ERG waveforms with characteristic a wave (i.e., the first negative wave reflecting the function of photoreceptor) and b wave (i.e., the following positive wave reflecting the activity of rod bipolar cells)[42]. The corresponding coefficient of variation (CV) of the amplitudes and implicit times of the ERG waveforms is summarized in Fig. 4e. It is clear that the corneal sensor provided the highest signal amplitude of the a and b waves, while the measurements were most consistent (CV < 6.4%) without noticeable blinking artifacts (Supplementary Table S2). These results suggest that the corneal sensor was more intimately interfaced with the corneal surface than other devices, despite blinking and eye movements. The ERG-Jet lens also exhibited high signal amplitude, but the measurements were unstable (CV > 15.7% in amplitudes) due to the non-conformal contact to the eye. In addition, the required use of corneal anesthetic agents for the ERG-Jet lens might provide a potential risk of reducing the amplitudes and prolonging the implicit times[22,43]. As expected, the DTL fiber showed the lowest amplitude signals due to the far distance from the cornea, and the measurements were prone to variations by blinking artifacts (CV > 10% in amplitudes).

The visual acuity of the participant (20/15) remained unchanged prior to and following testing with each of the devices. Upon slit lamp biomicroscopic examination with sodium fluorescein installation prior to testing, the participant had only minor, nonclinically significant superficial punctate staining (common to minor end of day dryness with SCL wear). Following wear of the corneal sensor for >1 h, the punctate staining had resolved. However, a nonclinically significant minor indentation arcuate staining (~1 mm in extent) was present in one quadrant of the superior cornea, which mirrored the location and orientation of the serpentine trace of the corneal sensor. This staining resolved within 2 h post lens removal.

**Standard full-field ERG recording in the human eye**. The International Society for Clinical Electrophysiology of Vision (ISCEV) standard for full-field clinical ERG signals specifies six responses based on the adaptation state of human eyes and the flash strength: (1) Dark-adapted 0.01 ERG (rod ERG), (2) Dark-adapted 3.0 ERG (combined rod–cone standard flash ERG), (3) Dark-adapted 3.0 oscillatory potentials, (4) Dark-adapted 10.0 ERG (strong flash ERG), (5) Light-adapted 3.0 ERG (standard flash "cone" ERG), and (6) Light-adapted 30 Hz flicker ERG[44]. The ISCEV encourages the use of additional ERG protocols for testing beyond the minimum standard for clinical ERG signals, which are abbreviated as Dark 0.01, Dark 3.0, Dark OP, Dark 10.0, Light 3.0, and Light flicker, respectively. The standard ERG waveforms with characteristic amplitudes and implicit times are noted in Supplementary Figure S14.

Figure 5a shows representative measurement results of the standard full-field ERG signals that were sequentially measured using the corneal sensor (red lines) by comparison with the ERG-Jet lens (blue lines) and the DTL fiber (green lines)[32]. The ERG recordings by using the corneal sensor showed consistent results of the highest amplitudes throughout the entire testing period of each session (typically <30 min), while providing better comfort

of wear than other devices. The detailed analyses of each full-field ERG waveform is shown in the "Methods" section. Figure 5b shows a summary of the average amplitudes (left column) and implicit times (right column) obtained from at least eight repeated recordings for each ERG protocol. The results obtained from a one-way analysis of variance (ANOVA) confirm that the amplitudes of the corneal sensor were significantly higher than those of other devices ($p < 0.0001$), while the implicit times remained statistically unchanged.

## Discussion

The outcomes reported here establish an innovative platform technology that enables turning common commercially available disposable SCLs into a functional corneal sensor tailored for ophthalmic ERG testing in human eyes. The resulting device is thin and deformable, and can be printed on a SCL without substantially altering the intrinsic lens properties in terms of biocompatibility, softness, oxygen permeability, transparency, wettability, and ergonomic curvature. A strategy that utilizes an electrochemical anchoring of the device to SCLs provides a guideline to enhance the mechanical robustness and chemical stability, in order to meet the requirements for lens fitting, handling, cleaning, and disinfection. The use of commercially available SCLs allows the device to form a conformal, seamless contact to a variety of corneal shapes, and therefore provides superior comfortability and on-eye safety compared to current clinical standards (e.g., the ERG-Jet lens and the Burian-Allen lens). The findings from the first-in-human validation study confirm the capability of the device in the high-fidelity recording of standard full-field ERG signals with a high signal-to-noise ratio. Importantly, the ERG recording is accomplished in a manner that allows for natural blinking and eye movements, without topical anesthesia or a speculum that is typically used in current ophthalmic examinations despite its adverse effects. Moreover, the fabrication of the device involves the use of a well-established DIW method that may facilitate the exploration of high-throughput batch production, potentially making the final product disposable and affordable for widespread adoption in the future. This work produces a first-of-its-kind corneal sensor platform that may be also tailored for a broad range of ophthalmic and optometric clinical needs such as the continuous monitoring of IOP and eye movement[45–47].

## Methods

**Automated DIW process**. A glass substrate (i.e., a plain microscope slide, Dow Corning) was cleaned in a bath of acetone and isopropyl alcohol with sonication for 30 min each, followed by exposure to ultraviolet (UV)-ozone for 10 min. A water-soluble PVA (10 wt% of Mowiol 4-88 in DI water) was spun coat on the glass substrate at 1000 r.p.m. for 30 s to serve as a sacrificial layer, and then baked at 100 °C for 30 min. In parallel, the formulated PDMS ink was prepared by mixing the base solution (DOWSIL SE1700, Sylgard 184, Dow Corning) and curing agent with a weight ratio of 1:1:0.2. The formulated AgSEBS ink was prepared by dissolving 1.5 g of SEBS (H1221, Asahi Kasei) in 1.5 g of tetrahydrofuran (Sigma-Aldrich) and 4 g of 1,2-dichlorobenzene (Sigma-Aldrich), and mixing 8 g of Ag flakes (2–5 μm, Inframat Advanced Materials) with a planetary centrifugal mixer (Thinky, ARE-310). Direct writing of the PDMS ink was carried out on the glass substrate coated with the PVA sacrificial layer to define the bottom encapsulation layer. For this, a stainless-steel tip (Nordson EFD) with an inner diameter of 200 μm was used by applying the pneumatic pressure and printing speed at 25 psi and 8 mm s$^{-1}$, respectively. Following thermal annealing of the printed PDMS ink at 70 °C for 30 min, another direct writing of the AgSEBS ink was carried out to define the conduction paths. For this, a stainless-steel tip with an inner diameter of 150 μm was used by applying the pneumatic pressure and printing speed at 35 psi and 8 mm s$^{-1}$, respectively. The entire structure was then heated at 70 °C for 1 h. Finally, one more direct writing of the PDMS ink was carried out to define the top encapsulation layer, followed by annealing at 70 °C for 30 min. The entire structure was then immersed in DI water to dissolve the underneath PVA layer, allowing the corneal sensor to be released from the glass substrate. The conduction paths (i.e., AgSEBS) within the corneal sensor was then plated with Au in a 24K pure gold plating solution for 30 s by using a general plating kit (Gold Plating Services) at the applied voltage of 3 V. After rinsing with DI

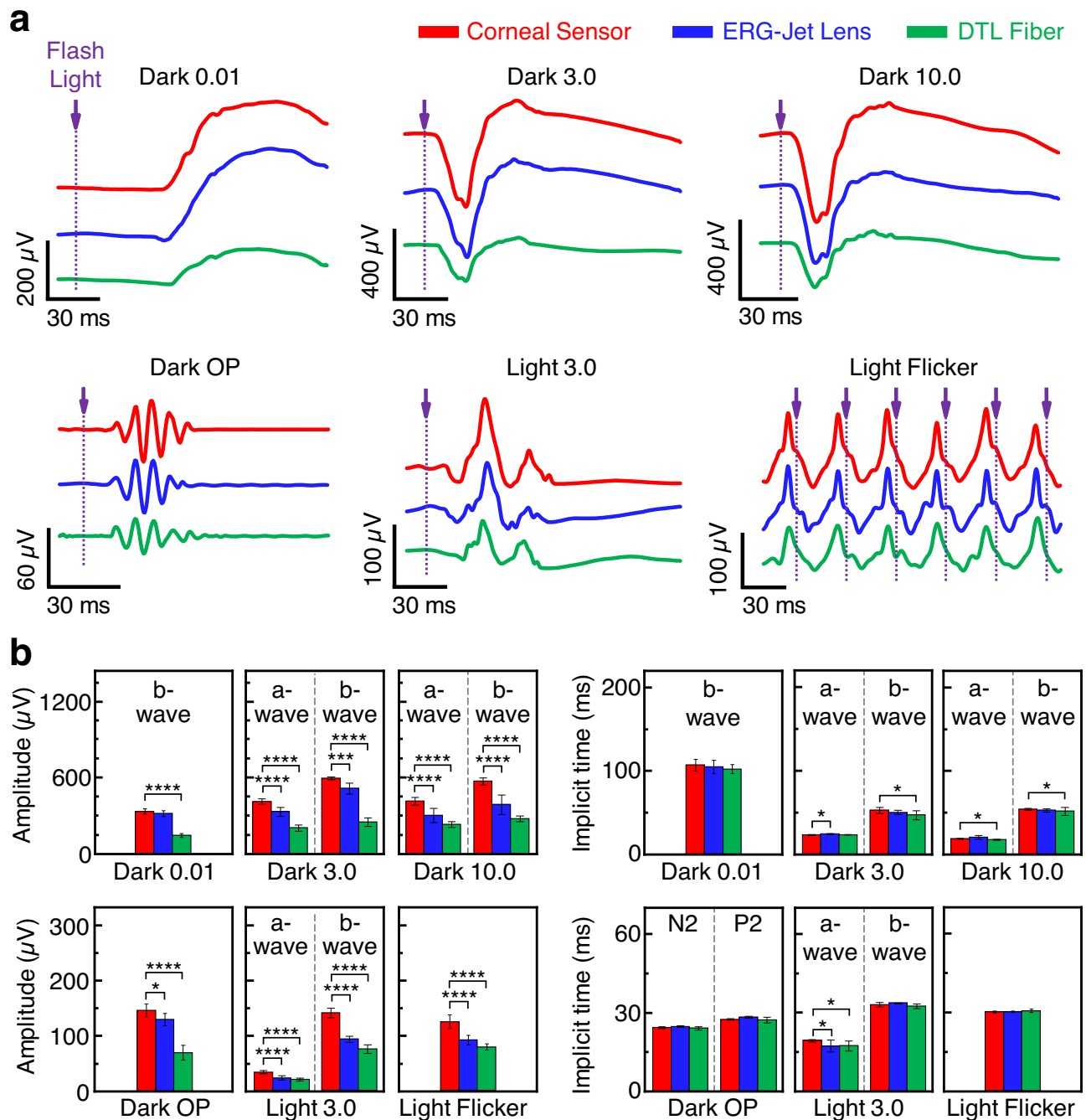

**Fig. 5 ISCEV standard full-field ERG recordings in a human eye. a** ISCEV standard full-field ERG signals acquired from the corneal sensor (red lines) by comparisons with those obtained using the ERG-Jet lens (blue lines) and DTL fiber (green lines). **b** Summary of average amplitudes (left column) and implicit times (right column) extracted from each ERG protocol. The number of recordings for each protocol was 8 (Dark 0.01), 10 (Dark 3.0), 10 (Dark 10.0), 8 (Dark OP), 8 (Light 3.0), and 9 (Light Flicker), respectively. For data analysis, a one-way analysis of variance (ANOVA) method with the Tukey's post hoc test was used. Significance was set at ****$p < 0.0001$, ***$p < 0.001$, and *$p < 0.1$.

water, the Au-coated corneal sensor was then transferred to the inner surface of a SCL, while the elastomeric connection wire was penetrated out through the SCL on the edge. Different types of commercially available disposable SCLs (Frequency 55 from CooperVision, ACUVUE Oasys from Johnson & Johnson, and Biotrue OneDay from Bausch & Lomb) were tested in this study.

**Electrochemical printing process.** The as-printed corneal sensor was immersed in DI water containing 0.02 M EDOT (Sigma-Aldrich) and 0.1 M sodium $p$-toluenesulfonate salt (Sigma-Aldrich) for 30 min, while its connection wire was connected to a working electrode. The electrochemical polymerization of PEDOT was subsequently carried out with a platinum (Pt) counter electrode and Ag/AgCl reference electrode by applying the bias voltage at 1 V for 4 min. The complete

device was then immersed in DI water for 30 min, followed by thorough rinsing with a lens saline solution (Sensitive Eyes, Bausch & Lomb) to remove residual EDOT and salts. Furthermore, the device was immersed in a cleansing kit containing a 3% $H_2O_2$ formula (ClearCare®, Alcon) overnight for disinfection. The surface topology and thickness of the device were characterized using a P-7 surface profilometer (KLA-Tencor) and SEM (S-4800, Hitachi), respectively.

**Monitoring of electrochemical impedance.** The electrochemical impedance of the device was measured using a VersaSTAT 3 potentiostat analyzer (Princeton Applied Research) with a standard three-electrodes configuration in a PBS solution (pH = 7.4) at room temperature. A LowProfile platinum electrode (PINE Research) and LowProfile Ag/AgCl electrode (PINE Research) were used to serve as the

counter and reference electrode, respectively. A 10 mV root-mean-square AC voltage was applied at varied frequencies ranging from 0.1 Hz to 100 kHz during these measurements. To characterize the chemical stability, the device was stored in either lens cleaning solution, PBS, or an artificial tear solution. Prior to each test, the device was rinsed with PBS for 1 min and immersed in a new PBS for 30 min.

**Electrical and electrochemical measurements under repeated stretching**. The device was loaded on a chuck of a tensile testing machine (ESM303, Mark-10) and then stretched to a prescribed stretching ratio. The device was stretched and released at the elongation rate of 10–20% per minute, while its resistance was simultaneously monitored using a source meter (Keithley 2400). The connection wire was stretched and released at the elongation rate of 100% per minute. During these tests, the electrochemical impedance of the device was also monitored at every 150 cycles.

**Cell compatibility evaluation**. The device was sterilized with an ethanol–DI water mixture (70:30 v/v) for 30 min, rinsed with Dulbecco's PBS (Gibco), and dehydrated with UV irradiation for 1 h. The sterilized device was then placed inside a 24-well plate on a concave side facing upwards. HCEpiCs (MilliporeSigma) with a density of $1 \times 10^5$/well were seeded in a cell media (EpiGRO™ Human Ocular Epithelia Complete Media, MilliporeSigma) for 12 h, and subsequently incubated in a humid incubator maintained at 37 °C with 5% $CO_2$ for 24 h. A 3-(4,5-dimethylthiazol-2-yl)-2,5-diphenyltetrazolium bromide (MilliporeSigma) reagent was added and incubated for 3 h. Following removal of the cell media, the cells were lysed with dimethylsulfoxide (ATTC). The absorbance of each well was measured using a microplate reader (Synergy™ NEO, BioTek) at the wavelength of 575 nm. The statistical analysis was carried out using a one-way ANOVA method with the Tukey's post hoc test implemented in the Origin software (OriginLab) and are expressed as averages ± s.e.m. ($n = 5$)

**FEA analysis**. The FEA analysis was conducted through the ABAQUS/Standard package. In the FEA model, the corneal sensor was bonded to the SCL to mimic their monolithic integration. The mechanical modulus ($E$) and Poisson's ratio of the corneal sensor and the SCL was 810 kPa and 0.4 and 14 MPa and 0.1, respectively. C3D4 elements were employed and mesh refinements in the corneal sensor were confirmed to capture the local stress concentration. For the flipping loading condition, only in-plane deformation at the edge of the corneal sensor was allowed, and a vertical force was applied at the bottom. For the folding loading condition, two rigid plates in contact with the corneal sensor were utilized and moved to achieve complete folding. For the stretching loading condition, the corneal sensor was initially clamped by two rigid plates at both ends and then was stretched under a quasi-static loading. For the expanding loading condition, a rigid circular plate was bonded at the edge of the corneal sensor and expanded uniformly in a controlled expanding ratio.

**Measurement of standard full-field ERG signals**. The measurement of the full-field ERG signals was conducted using an amplifier (PL3516, ADINSTRUMENTS) at a sampling rate and resolution of 2 kHz and 0.1 μV, respectively. All the tests on the human subject were conducted in a university ophthalmic clinic in accordance with Good Clinical Practice standards, university regulations, and institutional review board review. The authors affirm that human research participants provided informed consent for publication of the images in the Fig. 4a, b. The corneal sensor and DTL fiber were worn on the left eye of the participant before dark adaptation, while the ERG-Jet lens was necessarily worn after dark adaptation under a dim red light in order to minimize the wearing time due to corneal irritations. The ERG signals (Dark 0.01, Dark 3.0, and Dark 10.0) were measured under a light stimulus for 2 ms with the intensity of 0.01, 3.0, and 10 cd·s m$^{-2}$, respectively. Under a dim light of 0.01 cd·s m$^{-2}$ for the Dark 0.01, slowly responding b waves were observed due to the response time of a rod-driven on-bipolar cell in the retina. For the Dark 3.0, the a wave appeared before the b wave, while the amplitude of the b wave was increased with the faster response of the rod- and cone-driven bipolar cell activity. For the Dark 10.0, the a waves more clearly appeared relative to the Dark 3.0. The bandpass filters for the Dark 0.01, Dark 3.0, and Dark 10.0 were ranged from 0.3 to 300 Hz. A different bandpass filter (75–300 Hz) was used for the Dark OP. For the Light 3.0 and Light Flicker, the light adaptation was carried out at 30 cd m$^{-2}$ for 10 min. The Light 3.0 was measured under the background light of 30 cd m$^{-2}$ and single light stimulus for 2 ms at the intensity of 3.0 cd·s m$^{-2}$. The Light Flicker was obtained with multiple light stimuli at 30 Hz. The applied bandpass filter was ranged from 0.3 to 300 Hz. The statistical analysis for the amplitude and implicit time of the ERG signals was carried out using a one-way ANOVA method with the Tukey's post hoc test implemented in the Origin software (OriginLab).

**Ocular coherence tomography**. The OCT images were acquired prior to and during the application of the device using a Zeiss Visante OCT (Carl Zeiss). The data acquisition was made in corneal high-resolution mode at 16 meridians (e.g., slices).

**Slit lamp micrograph**. The slit lamp biomicroscopic examinations with and without sodium fluorescein installation were performed prior to each testing, with the device in place, and following testing. Magnifications ranging from ×10 to ×16 were used.

## Data availability
The data that support the findings of this study are available from the corresponding authors upon reasonable request.

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

## Acknowledgements

We thank Dr. Arthur Bradley for his insightful comments and suggestions in the early stage of this project. All studies on a human subject were conducted in accordance with the ethical guidelines of the National Institute of Health (NIH) and with the approval of the Institutional Review Board (IRB) of the Indiana University, Bloomington. This project was supported by the National Science Foundation (NSF) (CMMI-1928784). The work of H.Z. and B.X. was kindly provided by the NSF grant (CMMI-1928788); The work of H.J.K. and B.W.B. was kindly provided by the Air Force Office of Scientific Research (AFOSR) through the Organic Materials Chemistry Program (FA9550-19-1-0271).

## Author contributions

C.H.L., B.W.B. and P.K. conceived of the overall research goals and aims; K.K., H.J.K., W.P., M.K.K., B.K., H.P., B.W.B. and C.H.L. were involved in the manufacturing of the devices; K.K., P.K. and C.H.L. were responsible for the overall engineering design of the devices; K.K., H.J.K., D.M., M.K.K., H.P., P.K. and C.H.L. were responsible for the collection and analysis of human data; H.Z. and B.X. were responsible for the mechanics analysis; K.K., H.J.K., H.Z., W.P., B.X., P.K., B.W.B. and C.H.L. were responsible for original drafting of the manuscript. All authors contributed to the writing and editing of the manuscript.

## Competing interests

The authors declare no competing interests.
