## [Peer Review File · Nature Communications]

Reviewer #1 (Remarks to the Author):

The authors have presented an innovative application on electroretinogram which I find unusual and novel. The paper is thorough and presents a nearly pragmatic approach. I would like to recommend the authors conduct two additional testings to provide insight on failure of the device. As an example, what environmental condition will lead it to the device failure. It can be temperature, humidity, etc. Another aspect is what mechanical deformation will make the device non-functioning anymore.

Reviewer #2 (Remarks to the Author):

In the manuscript entitled "All-Printed Stretchable Corneal Sensor on Soft Contact Lenses for Noninvasive and Painless Electrodiagnosis", Kim et al. demonstrated the fabrication of a corneal sensor by printing a stretchable electrode with a serpentine structure on the commercial soft contact lens. The authors' main contributions include (1) developing a corneal sensor with an electrochemical anchoring between the corneal electrode and soft contact lens; and (2) demonstrating the recording of standard full-field electroretinogram (ERG), which was comparable to the clinical standard cases such as ERG-Jet lens and DTL fiber. Unfortunately, the level of innovation and completeness of this manuscript is not significant enough to be published in Nature Communications due to the following reason:

Firstly, the device reported in this work does not present enough novelty. The authors have developed a serpentine-structured corneal electrode using a conventional dispensing method with silver flake-filled polystyrene block copolymers (i.e. polystyrene-b-poly(ethylene-co-butylene)-b-polystyrene). However, this serpentine geometry of the electrode is already well established in the field of stretchable electronics. In addition, the design of this corneal sensor and its recording mechanism are identical to the conventional gold-standard method [Doc. Ophthalmol. 1999, 98(3), 233; Acta Ophthalmol. Scand. 2001, 79(5), 497; Doc. Ophthalmol. 2004, 108(1), 77]. Similar to the standard device, this sensor in this manuscript still requires the ground electrode and reference electrode which need to be attached to the human skin near the eye with an entirely wiring form. Thus, the corneal sensor in this manuscript does not present enough novelty.

Secondly, in spite of the softness of contact lens and the stretchability of corneal electrode, this sensor is not imperative to comfortable electrodiagnosis. Previous contact-lens devices for the measurement of ERG signals require lid holders [Doc. Ophthalmol. 2015, 130 (1), 1]. For example, in the case of ERG-Jet, bumps are required to prevent the movements of the eye and electrode which can cause potential artifacts, constricting the accuracy of ERG recording. Although the authors have highlighted the negligible use of a speculum, the wired form of this corneal sensor in this work still needs to address this issue regarding the movements of the subject's eyelid, as ERG signals can be contaminated by artifacts related to the eye blinking motions.

Thirdly, many recent reports describe "wireless" functions of smart contact lenses. However, the "wired" device in this manuscript ultimately constrains the user's behavior and degrades the signal quality due to the eye blinking motions that touch the wire, which is not advanced compared to the wireless smart lenses.

Lastly, the experimental data is not sufficiently provided. For example:

1. The authors tested only one human subject, which is insufficient to provide representative comparisons among clinical standards (such as ERG-Jet and DTL fiber) by considering deviations of electrophysiological signals within each human subject. This is especially important because the corneal signal varies with the size and shape of the human eyeball.

2. Burian-Allen ERG electrode can provide the highest accuracy in electrodiagnostic eye tests. Although the authors compared their sensor characteristics with the cases of ERG-Jet and DTL fiber, the comparison with Burian-Allen electrode is missing. Due to those insufficient originality and imperativeness, this manuscript is unsuitable for publication in Nature Communications.

Reviewer #3 (Remarks to the Author):

The authors introduce a contact lens with embedded electronics capable of electroretinography (ERG). The contact lens contains a printed corneal sensor composed of PEDOT-encapsulated, electrically conductive AgSEBS+PDMS traces that are soft and stretchable. The authors perform mechanical and chemical characterization of the lens, including cytotoxicity measurements. Lastly, ERG recordings obtained with the printed contact lens are compared with those from a “gold standard” ERG-Jet lens.

Overall, this appears to be complete and original work that is suited for publication in Nature Communications. My only recommendation is to include a reference to the following paper, as it also presents a method for incorporating stretchable electronics into a soft contact lens:

Vásquez Quintero, A., Verplancke, R., De Smet, H. and Vanfleteren, J., 2017. Stretchable electronic platform for soft and smart contact lens applications. *Advanced Materials Technologies*, 2(8), p.1700073.

(I was not involved with this paper and have no connection to the authors.)

RESPONSE TO REVIEWER 1

We thank the reviewer for the highly favorable comments such as “The authors have presented an innovative application on electroretinogram which I find unusual and novel. The paper is thorough and presents a nearly pragmatic approach” and the recommendation for publication in this journal.

Reviewer’s Comment: I would like to recommend the authors conduct two additional testings to provide insight on failure of the device. As an example, what environmental condition will lead it to the device failure. It can be temperature, humidity, etc. Another aspect is what mechanical deformation will make the device non-functioning anymore.

Our Response and Revision: We thank the reviewer for this recommendation. In the revised manuscript, we show that the electrochemical impedance of the device in a phosphate buffer saline (pH = 7.4 at 37.5°C) was well-maintained at $14.7 \Omega \pm 3.9 \Omega$ with a frequency of 100 Hz against (1) a temperature cycling between 30°C and 80°C and (2) multiple dehydrations in ambient condition for at least 5 hours each (Supplementary Figure S12). The results also showed a tendency for a slight decrease in the impedance at high temperatures ($> 60^\circ\text{C}$). Importantly, no mechanical failure was found under these extreme environmental conditions. We also added the following text in the revised manuscript, “The impedance was also well-maintained under other harsh environmental conditions such as a temperature cycling between 30°C and 80°C and multiple dehydrations in ambient condition for at least 5 hours each (Supplementary Figure S12). The impedance was slightly decreased at high temperature ($> 60^\circ\text{C}$).” on page 4.

Supplementary Figure S12. a, Electrochemical impedance of the corneal sensor against a temperature cycling between 30°C and 80°C. b, The corresponding results with the fixed frequency of 100 Hz. c, Electrochemical impedance of the corneal sensor against multiple dehydrations in ambient condition for at least 5 hours each. d, The corresponding results with the fixed frequency of 100 Hz. e, Optical images of the corneal sensor throughout a cycle of dehydration and rehydration.

In the revised manuscript, we also show that the device was stretched up to the maximum strain of nearly 100% without mechanical failure, whereas the soft contact lens itself was torn into two pieces (Supplementary Figure S5). The corneal sensor was intact even after the contact lens was torn apart, confirming that the maximum stretchability of the device is determined by the soft contact lens. We also added the following sentence in the revised manuscript, “For instance, the corneal sensor was stretched without failure even after the SCL was torn into two pieces at the maximum strain of ~100% (Supplementary Figure S5).” on page 3.

Supplementary Figure S5. A series of photographs for the corneal sensor under stretching until it reaches the failure point.

RESPONSE TO REVIEWER 2

We thank the reviewer for the constructive feedback and comments. We believe that the changes in the revised manuscript address all the comments.

Reviewer's Comment #1: Firstly, the device reported in this work does not present enough novelty. The authors have developed a serpentine-structured corneal electrode using a conventional dispensing method with silver flake-filled polystyrene block copolymers (i.e. polystyrene-*b*-poly(ethylene-co-butylene)-*b*-polystyrene). However, this serpentine geometry of the electrode is already well established in the field of stretchable electronics. In addition, the design of this corneal sensor and its recording mechanism are identical to the conventional gold-standard method [Doc. Ophthalmol. 1999, 98(3), 233; Acta Ophthalmol. Scand. 2001, 79(5), 497; Doc. Ophthalmol. 2004, 108(1), 77]. Similar to the standard device, this sensor in this manuscript still requires the ground electrode and reference electrode which need to be attached to the human skin near the eye with an entirely wiring form. Thus, the corneal sensor in this manuscript does not present enough novelty.

Our Response and Revision: In the revised manuscript, we further highlighted the novelty of this work in more detail. To the best of our knowledge, this work demonstrates a first-of-its-kind stretchable corneal sensor that is thinly printed on the surface of various commercial soft contact lenses. The novelty of this work is that the use of commercial soft contact lenses allowed the device to form a conformal, seamless contact to the cornea of human eyes. In turn, the device provides superior comfortability, comfort, and safety compared to clinical standards (e.g., the ERG-Jet lens and the Burian-Allen lens) that are made up of a bulky, thick, and rigid contact lens with non-optimal geometries. We confirmed the ability of the device in high-fidelity recording of human electroretinogram (ERG) responses in a non-invasive manner, and therefore, we can retain good comfort while eliminating the need of corneal anesthesia or use of a speculum; both of which are required for the gold-standard measurement. It should be noted that we employed an already well-established serpentine geometry in order to provide optimal stretchability of the device. The recording mechanism and electrode configuration were followed as suggested by the international society for clinical electrophysiology of vision (ISCEV) standard protocols. We also added the following sentences in the revised manuscript, "The use of commercially-available SCLs allows the device to form a conformal, seamless contact to the cornea of human eyes, and therefore provides superior comfortability and safety compared to current clinical standards (e.g., the ERG-Jet lens and the Burian-Allen lens). The findings from the first-in-human validation study confirm the ability of the device in high-fidelity recording of standard full-field ERG signals with high signal-to-noise ratio. Importantly, this is accomplished in a manner that allows for natural blinking and eye movements, without the need of topical anesthesia or use of a speculum." on page 7.

Reviewer's Comment #2: Secondly, in spite of the softness of contact lens and the stretchability of corneal electrode, this sensor is not imperative to comfortable electrodiagnosis. Previous contact-lens devices for the measurement of ERG signals require lid holders [Doc. Ophthalmol. 2015, 130 (1), 1]. For example, in the case of ERG-Jet, bumps are required to prevent the movements of the eye and electrode which can cause potential artifacts, constricting the accuracy of ERG recording. Although the authors have highlighted the negligible use of a speculum, the wired form of this corneal sensor in this work still needs to address this issue regarding the movements of the subject's eyelid, as ERG signals can be contaminated by artifacts related to the eye blinking motions.

Our Response and Revision: We thank the reviewer for this comment. The connection wire of our device is exceptionally soft [Young's modulus (E) = 420 kPa], thin (120 μm -thick), lightweight ($\sim 1.4 \text{ mg cm}^{-1}$), and stretchable (up to 350% without mechanical failure); thus providing minimal effect on motion artifacts. For instance, we showed that the electrical properties of this wire were retained through

more than 1,500 cycles of stretching at 50% and twisting at $1,440^\circ$ (Figure 2b-d). We also showed that the conformal contact of the device to the corneal surface of a human eye was well-maintained against repeated blinking and eye movements (Supplementary Movies S4 and S5). As a consequence, we demonstrated the reliable recording of ERG signals with high signal-to-noise ratios (Figure 5). On the other hand, the clinical standard device (e.g., the ERG-Jet lens) was systematically decentered by the movement of the relatively stiff ($E = 1.3 \text{ GPa}$), thick ($\sim 0.6 \text{ mm}$ -thick), and heavy ($\sim 8.6 \text{ mg cm}^{-1}$) connection cable made of polyvinyl chloride (PVC)-coated lead cable (Supplementary Movie S6), and thus, this device requires lid holders. To further evaluate the effect of motion artifact on signal quality, we show that the electrochemical impedance of our device in a phosphate buffer saline (pH = 7.4 at 37.5°C) remained sufficiently low at $18.2 \pm 3.8 \ \Omega$ even against tapping, swinging, and spinning of the connection wire (Supplementary Figure S7). We also added the following sentences in the revised manuscript, “The electrochemical impedance remained sufficiently low at $18.2 \pm 3.8 \ \Omega$ even against tapping, swinging, and spinning of the connection wire (Supplementary Figure S7), implying that the effect of motion artifacts on signal quality is insignificant.” on page 4 and “(iv) The external connection wire of the corneal sensor was thin ($120 \ \mu\text{m}$ -thick), lightweight (1.4 mg cm^{-1}), and sufficiently soft ($E = 420 \text{ kPa}$) enough to avoid any interruption from blinking and eye movements (Supplementary Movie S5). On the other hand, the polyvinyl chloride (PVC)-coated lead cable of the ERG-Jet lens was considerably thicker (0.6 mm -dia.), heavier (8.6 mg cm^{-1}), and stiffer ($E = 1.3 \text{ GPa}$), making it difficult to align the lens to the pupil center and capture consistent ERG signals (Supplementary Movie S6).” on page 6.

Supplementary Figure S7. Electrochemical impedance of the corneal sensor against tapping, swinging, and spinning of the connection wire, as compared to that under stationary condition.

Reviewer’s Comment #3: Thirdly, many recent reports describe “wireless” functions of smart contact lenses. However, the “wired” device in this manuscript ultimately constrains the user’s behavior and degrades the signal quality due to the eye blinking motions that touch the wire, which is not advanced compared to the wireless smart lenses.

Our Response and Revision: We thank the reviewer for this comment. The wireless function of smart contact lenses is designed for long-term (days) continuous monitoring of biosignals, which is unnecessary and has previously never been used for ERG recording because it typically requires no more than 30 minutes. It is also noted that the wirelessly-addressable smart contact lenses, such as the Sensimed Triggerfish[®], suffer from many side effects including conjunctival epithelial defects, conjunctival erythema, and pain due to the thick ($583 \ \mu\text{m}$ -thick) and rigid ($E > 130 \text{ GPa}$) nature of a silicon chip

[Dunbar GE, et al., Clin. Ophthalmol., 11, 875-882 (2017)]. We believe that the rigid form factor of currently-available chips precludes their application to ERG devices (or other corneal devices, for that matter). In fact, all existing corneal sensors for ERG recording demand a wire connection to an external unit allowing for data acquisition, light stimulation (Ganzfeld), and the real-time monitoring of the pupil size. This measurement setting is universal by complying with the ISCEV standard protocols. Importantly, even though it is wired, the connection wire of our device does not constrain the user's behavior and degrade the signal quality by motion artifacts (please see our response and revision for the reviewer's comment #2).

Reviewer's Comment #4-1: Lastly, the experimental data is not sufficiently provided. For example: 1. The authors tested only one human subject, which is insufficient to provide representative comparisons among clinical standards (such as ERG-Jet and DTL fiber) by considering deviations of electrophysiological signals within each human subject. This is especially important because the corneal signal varies with the size and shape of the human eyeball.

Our Response and Revision: We thank the reviewer for this comment. In this study, we compared the performance of our device with clinical standards (e.g., the ERG-Jet and the DTL fiber) on the same eye of a healthy human subject who had no history of ocular disease to eliminate the effect of different eye size and shape on ERG signals. To provide a statistically significant data set, we averaged the characteristic parameters (e.g., amplitude, implicit time, and variability) of each device that were obtained from at least 8 repeated recordings of each ERG protocol within the subject. These comparisons are consistent with previous studies [(10) Eye, 1993, 7, 169-171 & (11) Investigative Ophthalmology & Visual Science, 2017, 58, 4890]. Importantly, it should be noted that patient-to-patient variation is not a critical concern here. This is because the measurement of ERG signals, after all, requires no calibration among different subjects because the initial measurement data set is used as a reference baseline for each subject. To avoid any confusion, we added the following sentence in the revised manuscript, "...on the same eye of the participant, and thereby can eliminate the effect of different size and shape of human eyeballs on ERG signals (Figure 4b, middle and bottom panels, respectively).³²" on page 5. We also added the relevant references.

Reviewer's Comment #4-2: 2. Burian-Allen ERG electrode can provide the highest accuracy in electrodiagnostic eye tests. Although the authors compared their sensor characteristics with the cases of ERG-Jet and DTL fiber, the comparison with Burian-Allen electrode is missing.

Our Response and Revision: We thank the reviewer for this suggestion. The Burian-Allen ERG electrode is not easily tolerated even with topical anesthesia due to the bulky size of the built-in speculum [(12) Sains Malaysiana 43, 7, 1089–1094 (2014)]. It is also noted that the Burian-Allen ERG electrode is typically used on sedated patients in a specific clinical trial that demands a period of several hours, although a session of no more than 30 minutes is recommended. [(13) Coupland, S. 2006. "Electrodes for Visual Testing." In Principles and Practice of Clinical Electrophysiology of Vision, edited by John R. Heckenlively and Geoffrey B. Arden]. For these reasons, the Burian-Allen electrode is not commonly utilized, and the control measurements using it were not included in this study. To avoid any confusion, we added the following sentences in the revised manuscript, "However, the bulky size of the built-in speculum creates its own discomfort and thereby limits its use for children or adults with small eyelid fissures.¹²" on page 2, "Due to these reasons, these devices are only used in rare instance and on sedated patients.¹³" on page 2, and "In this study, the Burian-Allen ERG electrode was not included due to the high discomfort and low tolerance experienced by the participant." on page 5. We also added the relevant references.

RESPONSE TO REVIEWER 3

We thank the reviewer for the highly favorable comments such as “Overall, this appears to be complete and original work that is suited for publication in Nature Communications.” and the recommendation for publication in this journal.

Reviewer’s Comment: My only recommendation is to include a reference to the following paper, as it also presents a method for incorporating stretchable electronics into a soft contact lens: Vásquez Quintero, A., Verplancke, R., De Smet, H. and Vanfleteren, J., 2017. Stretchable electronic platform for soft and smart contact lens applications. *Advanced Materials Technologies*, 2(8), p.1700073.

Our Response and Revision: We thank the reviewer for this suggestion. In the revised manuscript, we added the reference paper [(16) *Advanced Materials Technologies*, 2017, 2, 8, 169-171].

Reviewer #1 (Remarks to the Author):

The authors have responded to my recommendations to meet the minimal level. However, with reviewer 2, I find the responses are not compelling enough.

Reviewer #2 (Remarks to the Author):

I read carefully the authors' responses to my comments and to those of the other two referees. Unfortunately, the authors have not addressed my comments satisfactorily, and I still believe that this manuscript is not suitable for publication in Nature Communications. The corneal sensor in this manuscript follows almost identical designs and operations of the conventional gold standard method, and the use of silver flake-filled block copolymers as well as the serpentine structure for stretchable soft devices do not present enough novelty. Also, I still think that the wireless ERG recording is important, and the authors should have compared the properties of this wire connection with the recent results of wireless smart contact lenses, rather than the relatively old case of triggerfish. Furthermore, the accuracy of this sensor should be compared with the method that can give the highest accuracy in electrodiagnostic eye tests (Burian-Allen ERG electrode). By considering deviations in the size and shape of the human eyeball, multiple human subjects should be tested as well.

POINT-BY-POINT RESPONSES TO THE REVIEWER

Thank you for sending us the reviewers' comments. In this response, we highlight the key advance of our work in view of the published literature and its technological viability for clinical applications, along with a detailed comparison of the advantages/disadvantages of our work over previously-reported wireless contact lens technologies. In addition, we also summarized the comments from the reviewer and our point-by-point responses to them. In a separate file, we intensively revised the manuscript (yellow-highlighted areas) to reflect our responses below.

(1) Key advance of our work and technological viability for clinical applications

Electroretinogram (ERG) examinations serve as routine clinical procedures in ophthalmology for the diagnosis and management of many ocular diseases. However, current clinical gold-standard method for measuring ERG responses in human eyes involves the use of an extremely thick, rigid contact lens sensor (e.g., the ERG-Jet lens). In addition, the contact lens sensor is only available in one shape and does not conform to any eye on which it is placed. Thereby, it typically requires the application of corneal anesthesia and a speculum that prevents eye closure. Our work creates a new contact lens sensor that is built upon a commercially-available (FDA-approved) soft contact lens, thereby leading to a significant improvement over the ERG-Jet lens. Being placed on a commercial soft contact lens, which conforms to an arbitrary corneal shape, our device provides unique capabilities to (1) capture high-fidelity ERG signals in human eyes without the use of corneal anesthesia or a speculum, (2) fit well for an arbitrary size or shape of human eyes, and (3) be less decentered on the eye by > 10 -fold compared to the ERG-Jet lens without scratching the corneal surface. As a consequence, our device allows for the non-invasive, painless, and accurate recording of full-field ERG responses in human eyes without side effects, which is also well-validated through comprehensive preclinical studies with experienced clinicians (Dr. Kollbaum & Dr. Meyer – two of the authors of the manuscript).

(2) Detailed comparison of the advantages/disadvantages of our work over current smart contact lenses

Recent technological advances have led to the development of industrial-grade smart contact lenses, such as the Sensimed TriggerFish lens and the Google smart contact lenses. These devices allow for (1) the continuous monitoring of intraocular pressure (IOP) or biomarkers (e.g., glucose) in tear films at corneal surface and (2) the wireless transmission of the data to the wearer through the use of an integrated circuit (IC) chip. However, the IC chip embedded in these devices is at least > 3 -fold thicker and $> 75,000$ -fold stiffer than a typical soft contact lens, which results in user discomfort and the risk of corneal hypoxia, especially if worn for a long period of time. Other side effects have been also reported, including foreign body sensation, eye pain, superficial punctate keratitis, corneal epithelial defects, and conjunctival erythema [US Ophthalmic Review, 6(1):10, 2013]. More recently, several ongoing research endeavors have helped enable the successful fabrication of a range of flexible sensors on a custom-built contact lens made from several polymers (e.g., hydrogel silicones, Parylene-C, or SU8 resins) and functional nanomaterials (e.g., graphene and metallic nanowires) [e.g., Sci. Adv., 6:eaba3252, 2020]. These newer devices have shown some initial success at the laboratory scale, but their practical application in human eyes remains impeded due to the lack of mechanical reliability (for lens handling, fitting, cleaning, and inadvertent eye rubbing), chemical stability (for long-term lens storage and multiple disinfection cycles), and oxygen transmissibility, among other reasons. Moreover, the custom-built contact lenses used in these devices still suffer from limited wettability and achieving ergonomic curvature, which may affect their long-term wearability for the human eye.

To address this critical opportunity and also further advance the technological viability for clinical applications, we developed an innovative strategy that involves the direct-in-writing (DIW) of a highly stretchable biosensor on various types of commercial disposable soft contact lenses. The resulting device offers excellent biocompatibility, softness [mechanical modulus (E) = 0.2-2 MPa], transparency ($\sim 100\%$), oxygen transmissibility (10-200 Dk/t), wettability (water content = 30-80%), and are also able to fit a variety of corneal shapes (8.3-9.0 mm base curve radii). As such, our device meets all the critical requirements for practical application in human eyes that are impossible to achieve using the recently-explored smart contact lenses. Moreover, our device is specifically tailored for the high-fidelity recording of ERG signals at the corneal surface of human eyes in a painless and unobtrusive manner that can eliminate the need for the use of topical corneal anesthesia or a speculum. In this design scheme, wireless ERG recording is unnecessary because the most of clinical ERG examinations are routine in-office procedures and typically occur within no more than 30 minutes in a clinic in the presence of a sophisticated light stimulator (e.g., a Ganzfeld stimulator). Instead, our device is connected to an external data acquisition system via a custom-built thin connection wire that is exceptionally stretchable (up to 350%) and lightweight ($\sim 1.4 \text{ mg cm}^{-1}$) to minimize the effect of blinking and eye rotational movements (e.g. on average around $\pm 4 \text{ mm}$) on signal quality. This connection wire is > 5 -fold thinner, > 6 -fold lighter, and $> 3,000$ -fold softer than a conventional lead wire that is also used for current gold-standard ERG sensors (e.g., the ERG-Jet lens).

Consequently, our device offers significant advantages over both the commercially-available clinical vision technologies and the recently-explored smart contact lenses, which makes our work novel from both an academic and

applied perspective. (1) Our device consists of intrinsically stretchable polymers of which the stacked layers remain at least 7-fold thinner, 2-fold softer, and 10-fold more stretchable compared to commercial soft contact lenses. Our device is also > 25-fold thinner, > 3-fold lighter, and > 2,000-fold softer than the ERG-Jet lens. (2) Our device is directly printed on various types of commercial soft contact lenses without substantially altering the intrinsic lens properties and therefore offers excellent wettability, biocompatibility, and oxygen transmissibility, compared to bare soft contact lenses. (3) Our device is monolithically bonded to commercial soft contact lenses through a novel electrochemical anchoring mechanism to provide sufficient mechanical and chemical reliability even under harsh environmental conditions including overstretching, a temperature cycling between 30 °C and 80 °C, and multiple dehydrations in ambient condition for at least 5 hours each. In preclinical tests, our device established a tight and conformal interface with the corneal anterior surface of human eyes with a comparable contact quality at the same level as bare soft contact lenses. These aspects allowed our device to provide significantly improved measurement accuracy and reliability, along with on-eye safety and patient comfort, compared to the ERG jet lens.

(3) Point-by-point responses to the reviewer's comments

Comment 1: The corneal sensor in this manuscript follows almost identical designs and operations of the conventional gold standard method, and the use of silver flake-filled block copolymers as well as the serpentine structure for stretchable soft devices do not present enough novelty.

Our response: As the reviewer noted, we employed the similar designs (e.g., serpentine layout), materials (e.g., silver flake-filled block copolymers), and operations of the conventional gold-standard smart contact lenses or other general types of stretchable biosensors. However, all these devices are fabricated on custom-built contact lenses or silicone elastomers and thereby suffer from limited oxygen transmissibility (leading to short-term wearability), wettability (leading to eye dryness and irritations), and softness and ergonomic curvature (leading to wearer discomfort) for the human eye. Consequently, their implementation in human eyes still remains impeded. Unlike these previous attempts, our device is built upon commercially-available (FDA-approved) soft contact lenses to meet all critical requirements for its practical application in human eyes with a great fit to an arbitrary corneal shape, as also described in detail above. This aspect allowed us to, for the first time, achieve the non-invasive and accurate recording of full-field ERG responses in human eyes without topical corneal anesthesia or a speculum that is typically used in current ophthalmic examinations despite its adverse effects.

Comment 2: Also, I still think that the wireless ERG recording is important, and the authors should have compared the properties of this wire connection with the recent results of wireless smart contact lenses, rather than the relatively old case of triggerfish.

Our response: The wireless ERG recording is unnecessary and, in fact, has never been used in clinical practice because the most of clinical ERG examinations are routine in-office procedures and typically occur for no longer than 30 minutes. Specifically, the patient is necessarily immobile during the ERG recording such that the controlled flux of light can be received into the eye of the patients from a sophisticated light stimulator (e.g., a Ganzfeld stimulator). The recently-explored wireless smart contact lenses [e.g., (1) *Sci. Adv.*, 6:eaba3252, 2020 and (2) *Nat. Comm.*, 8:14997, 2017], to which the reviewer is possibly referring, are completely different devices. That is, they are not capable of recording ERG signals, and their implementation in the long-term monitoring of human eyes still remains impeded due to the lack of mechanical and chemical reliability associated with on-eye safety and patient comfort

Comment 3: Furthermore, the accuracy of this sensor should be compared with the method that can give the highest accuracy in electrodiagnostic eye tests (Burian-Allen ERG electrode). By considering deviations in the size and shape of the human eyeball, multiple human subjects should be tested as well.

Our response: We reiterate that the Burian-Allen ERG electrode is used only on sedated patients due to severe eye discomfort for extremely rare clinical conditions that demand a long-term recording of ERG responses over several hours [Sains Malays, 43, 1089-1094, 2014]. In a typical clinical ERG examination session that occurs within no more than 30 minutes, the ERG-Jet lens is often used after topical corneal anesthesia. The ERG-Jet lens also provides high accuracy as comparable as the Burian-Allen ERG electrode. For these reasons, we used the ERG-Jet lens as a control group in our study. Moreover, the ERG signals are independent of the size and shape of the eye, which is also evidenced by the fact that currently-available commercial ERG sensors have only one size and shape to fit any patient (despite eye discomfort). According to the international society for clinical electrophysiology of vision (ISCEV) standard, the ERG measurement requires no calibration among different subjects because initial participant measurement data is used as a reference baseline.

Reviewer #1 (Remarks to the Author):

The authors have sufficiently revised the manuscript.

Reviewer #2 (Remarks to the Author):

Previously, I suggested the rejection of this manuscript for four main reasons: 1) the lack of novelty on the formation of this soft and stretchable device structure, 2) the wired ERG recording similar to previous operations, 3) no comparison of its accuracy with the most accurate method, 4) no data on multiple human subjects. Unfortunately, the authors have not provided any additional experimental results on these issues that I have raised last time, and their responses are less convincing. Therefore I still believe that this manuscript is not suitable for publication in Nature Communications.

Reviewer #3 (Remarks to the Author):

The authors have adequately addressed my comment.

POINT-BY-POINT RESPONSES TO THE REVIEWERS

We thank you and the reviewers for the careful reading and constructive feedback and comments. We were delighted to see that Reviewers 1 & 3 agreed with us and commented that our manuscript is adequately revised. We further revise our manuscript to address the comments from Reviewer 2 and editor.

Reviewer 2's Comments (Remarks to the Author): Previously, I suggested the rejection of this manuscript for four main reasons: 1) the lack of novelty on the formation of this soft and stretchable device structure, 2) the wired ERG recording similar to previous operations, 3) no comparison of its accuracy with the most accurate method, 4) no data on multiple human subjects. Unfortunately, the authors have not provided any additional experimental results on these issues that I have raised last time, and their responses are less convincing. Therefore, I still believe that this manuscript is not suitable for publication in Nature Communications.

Our Response: We note that Reviewers 1 & 3 found the responses convincing and the revision to manuscript sufficiently addressed the 4 aspects that Reviewer 2 had raised. Below we re-emphasize our responses to each comment of Reviewer 2. To further clarify these points, we further revised our manuscript as highlighted in yellow.

1) the lack of novelty on the formation of this soft and stretchable device structure: This work is novel as we, for the first time, achieved the non-invasive and accurate recording of full-field ERG responses in human eyes, which otherwise could not be achieved using any of existing contact lens-based sensor technologies.

2) the wired ERG recording similar to previous operations: Our ERG recording is unique because it does not require the use of topical corneal anesthesia or a speculum, which are typically used in current ophthalmic examinations despite their known adverse effects. We also reiterate that the wireless ERG recording is unnecessary and, in fact, has never been used in clinical practice because the most of clinical ERG examinations are routine in-office procedures and typically occur for no longer than 30 minutes.

3) no comparison of its accuracy with the most accurate method: In this work, we compared its accuracy with the most accurate method using a commercially-available gold standard contact lens-based corneal sensor (i.e., the ERG-Jet lens). The ERG-Jet lens provides the highest accuracy as comparable as the Burian-Allen ERG electrode.

4) no data on multiple human subjects: We reiterate that the ERG signals are independent of the size and shape of the eye, which is also evidenced by the fact that currently-available commercial ERG sensors have only one size and shape to fit any patient (despite eye discomfort). According to the international society for clinical electrophysiology of vision (ISCEV) standard, the ERG measurement requires no calibration among different subjects because initial participant measurement data is used as a reference baseline.